# Improved penicillin susceptibility of *Streptococcus pneumoniae* and increased penicillin consumption in Japan, 2013–18

Shinya Tsuzuki[1,2]*, Takayuki Akiyama[1], Nobuaki Matsunaga[1]*, Koji Yahara[3], Keigo Shibayama[3,4], Motoyuki Sugai[3], Norio Ohmagari[1,5]

**1** AMR Clinical Reference Center, National Center for Global Health and Medicine, Tokyo, Japan, **2** Faculty of Medicine and Health Sciences, University of Antwerp, Antwerp, Belgium, **3** Antimicrobial Resistance Research Center, National Institute of Infectious Diseases, Tokyo, Japan, **4** Department of Bacteriology II, National Institute of Infectious Diseases, Tokyo, Japan, **5** Disease Control and Prevention Center, National Center for Global Health and Medicine, Tokyo, Japan

* stsuzuki@hosp.ncgm.go.jp(ST); nomatsunaga@hosp.ncgm.go.jp(NM)

**Data Availability Statement:** Data are available within the Supporting Information file.

**Funding:** Our study was supported by a grant from the Ministry of Health, Labour and Welfare (Grant

## Abstract

### Objectives

To examine the association between penicillin susceptibility of *Streptococcus pneumoniae* and penicillin consumption in Japan.

### Methods

We used Japan Nosocomial Infection Surveillance data on the susceptibility of *S. pneumoniae* and sales data obtained from IQVIA Services Japan K.K. for penicillin consumption. We analysed both sets of data by decomposing them into seasonality and chronological trend components. The cross-correlation function was checked using Spearman's rank correlation coefficient to examine the correlation between susceptibility and consumption.

### Results

After adjusting for seasonality, the susceptibility of *S. pneumoniae* to penicillins gradually improved (55.7% in 2013 and 60.6% in 2018, respectively) and penicillin consumption increased during the same period (0.76 defined daily doses per 1,000 inhabitants per day [DID] in 2013, and 0.89 DID in 2018). The results showed positive cross-correlation (coefficient 0.801, *p*-value < 0.001). In contrast, cephalosporin consumption decreased (3.91 DID in 2013 and 3.19 DID in 2018) and showed negative cross-correlation with susceptibility of *S. pneumoniae* to penicillins (coefficient −0.981, *p*-value < 0.001).

### Conclusions

The rates of penicillin-susceptible *S. pneumoniae* isolates did not negatively correlate with penicillin consumption at the population level. Increased penicillin consumption might not impair the penicillin susceptibility of *S. pneumoniae*.

number 20HA2003, changed from the previous one) and Research Program on Emerging and Re-emerging Infectious Diseases from the Japan Agency for Medical Research and Development (AMED) under grant number JP19fk0108061.]. The funders had no role in study design, data collection and analysis, decision to publish, or preparation of the manuscript. No author received a salary from the funders.

**Competing interests:** The authors have declared that no competing interests exist.

## Introduction

Antimicrobial resistance (AMR) is currently one of the greatest threats to global health [1,2]. *Streptococcus pneumoniae* is one of the major targets of AMR surveillance [3–7] because both morbidity and mortality due to severe infections are substantial [8–10].

After publication of the Global Action Plan on Antimicrobial Resistance in 2015 [1], the Japanese government established its National Action Plan on Antimicrobial Resistance the following year [11]. This plan includes *S. pneumoniae* as one of the major target organisms and aims to improve the microbe's susceptibility to penicillin. In this plan, the susceptibility of *S. pneumoniae* to penicillins is described in terms of the minimum inhibitory concentration (MIC) threshold for pneumococcal meningitis defined by the Clinical Laboratory Standards Institute (CLSI) in 2015 [12]. According to this criterion, 48% of *S. pneumoniae* isolates in Japan were resistant to penicillins in 2014, and the Ministry of Health, Labour and Welfare (MHLW) of Japan set a goal to reduce the proportion of penicillin-resistant *S. pneumoniae* (PRSP) to below 15% by 2020 [11].

Because the guidelines published by the CLSI are now used in Japan, we used the interpretive MIC breakpoints for meningeal and non-meningeal isolates of *S. pneumoniae* according to the 2008 revision of CLSI guidelines [13]. However, these figures change considerably when the non-meningitis MIC threshold is applied. In 2014, 99.6% of *S. pneumoniae* isolates were susceptible to penicillins and in 2018, the number rose slightly to 99.7% [14]. Under these circumstances, it is difficult for us to find a substantial difference in the proportion of susceptible isolates before and after the action plan. If we used non-meningitis MIC, it would be difficult to find a time-series trend because most *S. pneumoniae* would be classified as "sensitive". However, the sample size would be too small if we used only cerebrospinal fluid specimens to monitor the penicillin susceptibility of *S. pneumoniae*. Considering this, it might be reasonable to apply the meningitis MIC threshold to all types of specimens (sputum, cerebrospinal fluid, etc.) to follow the trend of AMR in *S. pneumoniae*.

Nevertheless, evaluation by meningitis MIC threshold is not an established method for surveying trends in the penicillin susceptibility of *S. pneumoniae*. For instance, Torumkuney and colleagues evaluated the susceptibility of *S. pneumoniae* from respiratory tract specimens via both non-meningitis and meningitis MICs [3,15]. Karlowsky and colleagues reported $MIC_{90}$ as an indicator of resistance [6]. However, because $MIC_{90}$ to penicillins was 0.12 mg/L in most years, it is not a good alternative for monitoring the trend. Although the European Committee on Antimicrobial Susceptibility Testing (EUCAST) uses 0.06 mg/L as the threshold for penicillin susceptibility, it defined the threshold for penicillin resistance as 2.0 mg/L and did not mention the classification of isolates whose MICs were between 0.06 mg/L and 2.0 mg/L [16]. Additionally, EUCAST redefined its susceptibility testing categories (i.e. S, I, and R) in 2019 [17]. Under the new definition, *S. pneumoniae* isolates, which have MICs of between 0.06 mg/L and 2.0 mg/L, are evaluated as "Susceptible, increased exposure". Consequently, it is difficult to evaluate the penicillin susceptibility of *S. pneumoniae* according to EUCAST criteria only.

The Survey of Antibiotic Resistance has provided findings about AMR in various countries and reported the distribution among MICs including MIC50 and MIC90 [3,15,18–25]. According to these previous studies, *S. pneumoniae* seldom showed MICs greater than 2 mg/L to penicillins in most countries, although the distribution of MIC values differed in each country. In other words, meningitis MIC might be a more appropriate threshold for monitoring the antibiotic susceptibility of *S. pneumoniae*. However, non-meningitis MIC might be beneficial for clinical use because most community-acquired pneumococcal pneumonia can be treated with penicillin regardless of the *in vitro* MIC value [26,27].

Another concern is the recommendation of antibiotic prescriptions for *S. pneumoniae*. Needless to say, antibiotics should never be prescribed unless they are necessary, because the use of any type of antibiotic could exert selective pressure that could lead to AMR [28,29]. Many guidelines recommend the penicillin class of antibiotics as the first choice for *S. pneumoniae* infections due to its effectiveness and narrow spectrum [30–32]. However, it is unclear whether increased penicillin consumption at the population level could lead to reduced penicillin susceptibility of *S. pneumoniae*.

This study had two main objectives. The first was to assess the appropriateness of meningitis MIC (0.06 mg/L) as an indicator for the penicillin susceptibility of *S. pneumoniae*. The other was to evaluate the relationship between penicillin consumption at the population level and the penicillin susceptibility of *S. pneumoniae*.

## Materials and methods

### Data source

We used data collected by Japan Nosocomial Infections Surveillance (JANIS), which was organized by the Ministry of Health, Labour and Welfare [33,34]. The JANIS Clinical Laboratory module comprehensively collects all routine microbiological test results, including culture-positive and -negative results from over 2000 hospitals voluntarily participating in the surveillance, which account for a quarter of the approximately 8000 hospitals across Japan.

We extracted the monthly data for *S. pneumoniae* (including isolates from blood, cerebrospinal fluid, and respiratory tract specimens) and its susceptibility to antibiotics from January 2013 to December 2018 from the database of the JANIS Clinical Laboratory module. Patients were de-identified by each hospital before the data were submitted to JANIS. We included 636 facilities that continuously submitted their data to JANIS between 2013 and 2018. Approval for the extraction and use of the data was granted by the Ministry of Health, Labour and Welfare (0214–3).

To assess antibiotic consumption, we used monthly sales data collected by IQVIA Services Japan, which covers more than 99% of drug distribution among wholesalers in Japan. We calculated the monthly AMU using defined daily doses per 1,000 inhabitants per day (DID) from January 2013 to December 2018. We defined antimicrobials as J01 according to the ATC classification [35].

### Statistical analysis

We prepared monthly time-series data about the rate of penicillin-susceptible *S. pneumoniae* isolates and sales volume of antibiotics by class (penicillins, cephalosporins, and sum of all classes). First, we decomposed both datasets into a trend component and a seasonality component to evaluate the chronological trend by using locally-weighted scatterplot smoother (LOWESS) method. Next, we used Spearman's rank correlation test to examine the correlation between components. Additionally, we checked the correlation between susceptibility of *S. pneumoniae* to penicillins and antibiotic consumption. A *p* value < 0.05 was considered statistically significant. All analyses were conducted by R version 3.6.3 [36].

## Results

### Penicillin susceptibility of *S. pneumoniae* isolates

By applying non-meningitis MICs, the rates of penicillin-susceptible isolates were 97.8% in 2013 and 98.3% in 2018 (Fig 1). However, when applying meningitis MICs, the rates were

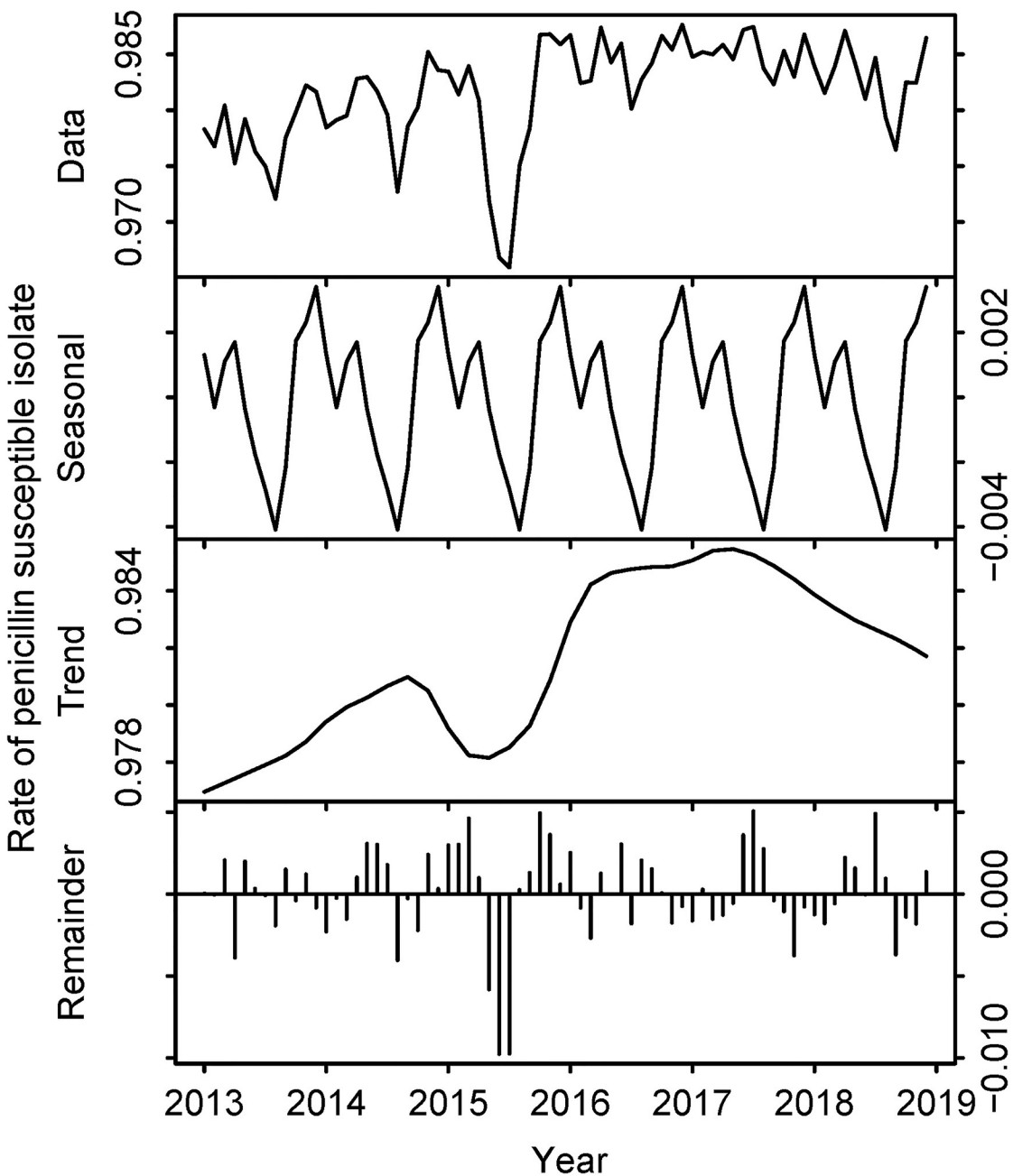

**Fig 1. Trend of penicillin susceptibility of *Streptococcus pneumoniae* in Japan, 2013–2018 (based on non-meningitis MICs).** The top row represents the raw data. The second row describes seasonality components. The third row describes trend components. The bottom row represents the remainder. Horizontal axes represent month and year. Vertical axes represent the rates of penicillin-susceptible *S. pneumoniae* isolates.

55.7% for 2013 and 60.6% for 2018 (Fig 2). The difference in the rates of susceptible isolates between the two cases is shown in Table 1 and Fig 3.

In Fig 2, the rate of susceptible isolates increased as time elapsed, after adjusting for seasonality. Fig 4 represent the distribution of MICs of *S. pneumoniae* isolates in each year.

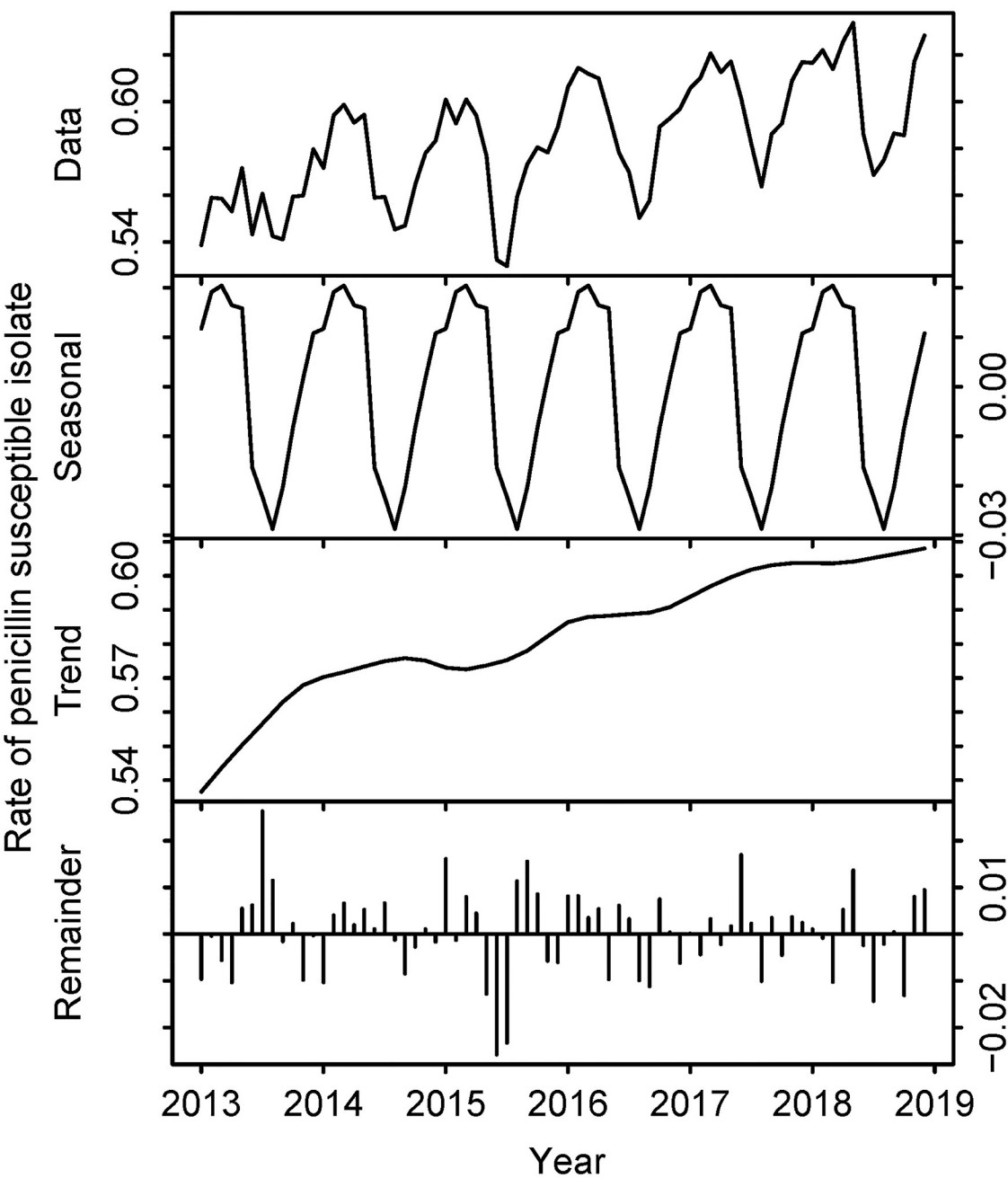

**Fig 2. Trend of penicillin susceptibility of *Streptococcus pneumoniae* in Japan, 2013–2018 (based on meningitis MICs).** The top row represents the raw data. The second row describes seasonality components. The third row describes trend components. The bottom row represents the remainder. Horizontal axes represent month and year. Vertical axes represent the rates of penicillin-susceptible *S. pneumoniae* isolates.

## Sales volume of antibiotics

While the sales volume of penicillins increased gradually (0.76 DID in 2013 to 0.89 DID in 2018), sales decreased for both cephalosporins (3.91 DID in 2013 to 3.19 DID in 2018) and total antibiotics (14.52 DID in 2013 to 12.91 DID in 2018) during the same period. The details are shown in Table 2 and Figs 5–8.

**Table 1.  Rates of penicillin-susceptible *Streptococcus pneumoniae* isolates in Japan, 2013–2018.**

|  | Meningitis MICs (≤0.06 mg/L) | Non-meningitis MICs (≤2 mg/L) |
|---|---|---|
| **2013** | 55.7% | 97.8% |
| **2014** | 57.6% | 98.1% |
| **2015** | 57.7% | 98.0% |
| **2016** | 59.0% | 98.5% |
| **2017** | 60.2% | 98.5% |
| **2018** | 60.6% | 98.3% |

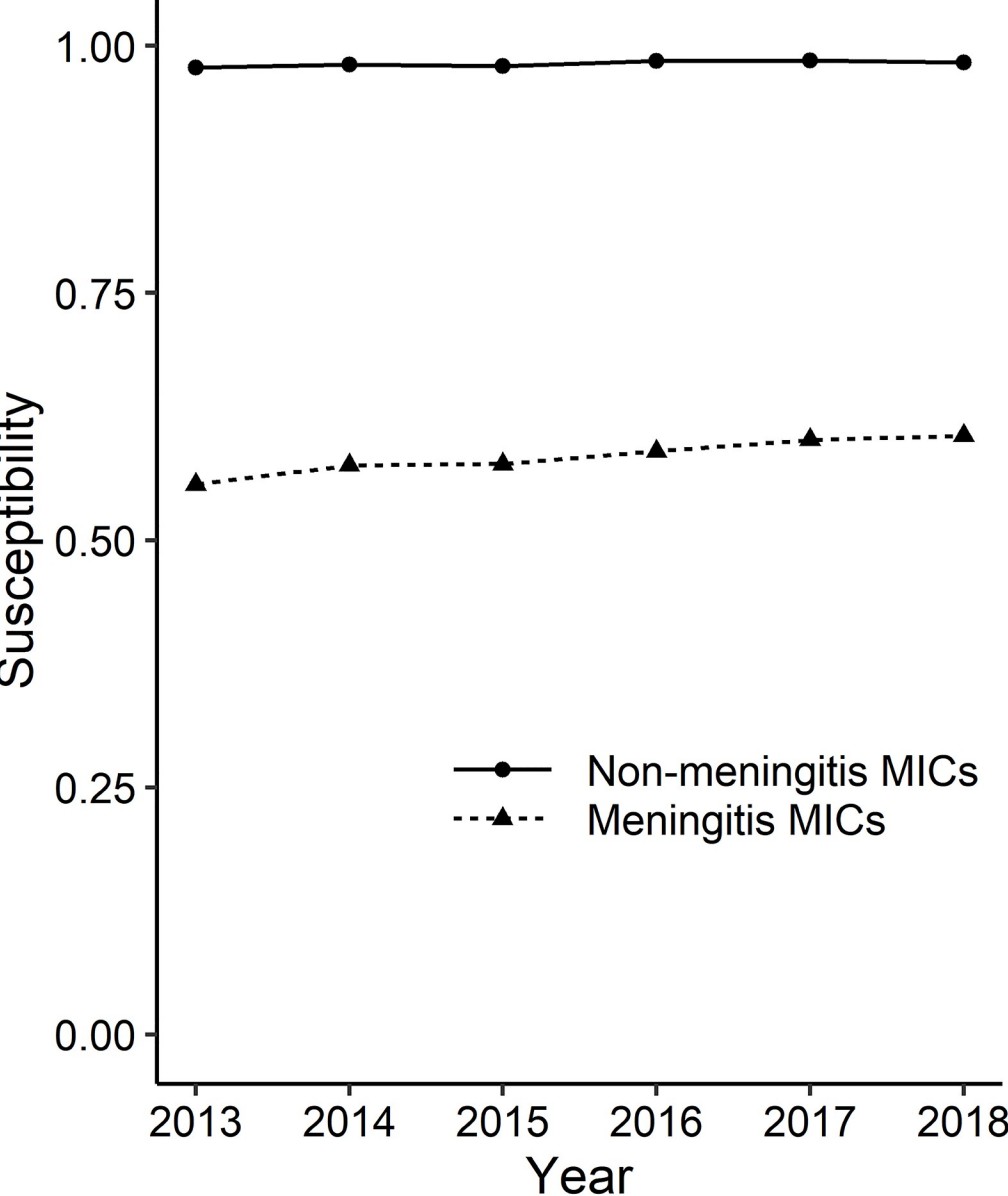

**Fig 3.  Annual change in penicillin susceptibility of *Streptococcus pneumoniae* in Japan, 2013–2018 (based on meningitis and non-meningitis MICs).** The solid line with circles represents the results based on non-meningitis MICs and the dashed line with triangles represents the results based on meningitis MICs.

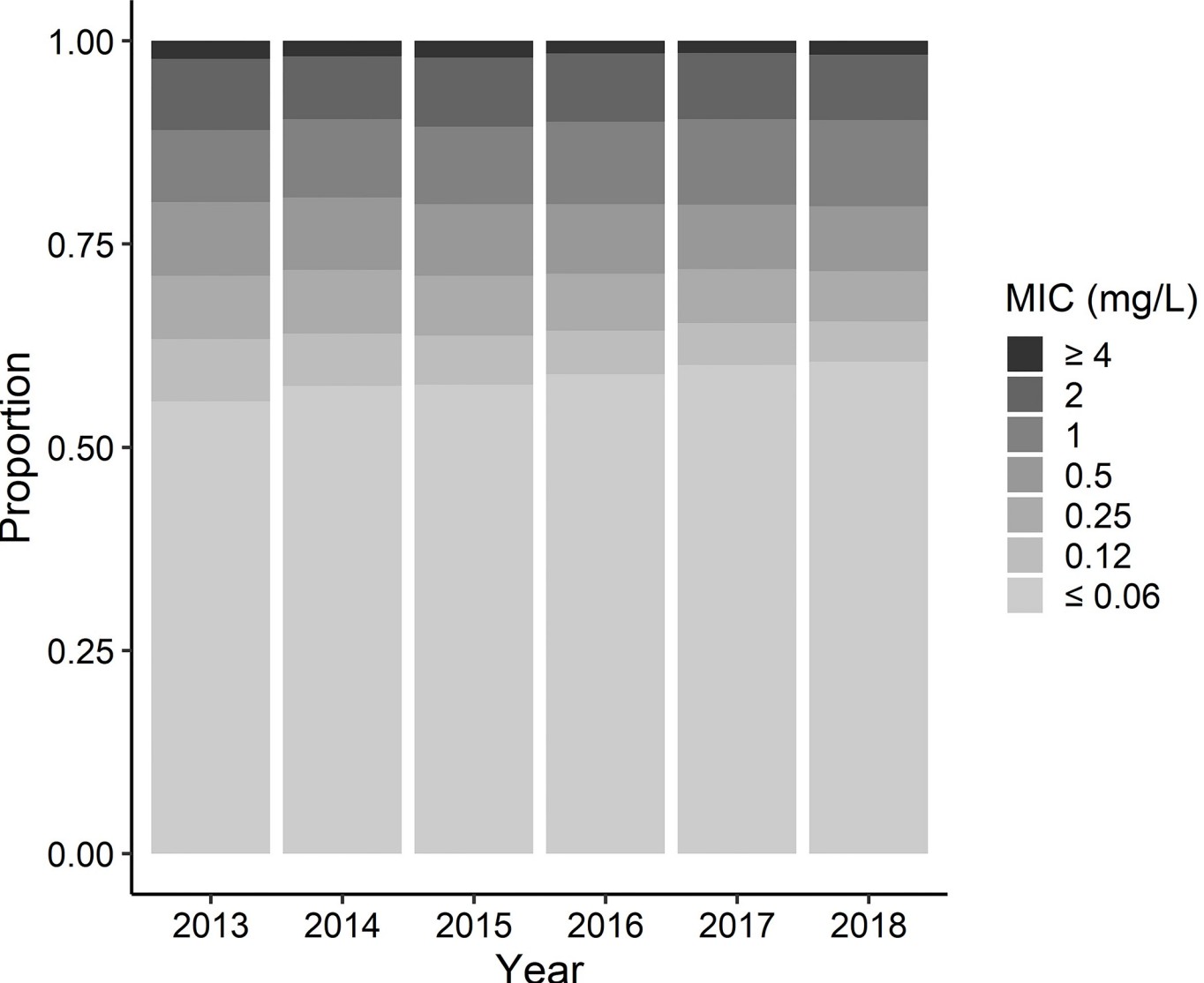

**Fig 4. Annual change in the distribution of MIC values for penicillin among *Streptococcus pneumoniae* isolates.** The darker areas represent the proportion of strains which showed higher MIC values.

**Table 2. Annual sales volume of antibiotics in Japan, 2013–2018*.**

|  | Cephalosporins | Penicillins | Total |
|---|---|---|---|
| **2013** | 3.91 | 0.76 | 14.52 |
| **2014** | 3.78 | 0.78 | 14.07 |
| **2015** | 3.82 | 0.85 | 14.23 |
| **2016** | 3.68 | 0.84 | 14.15 |
| **2017** | 3.43 | 0.83 | 13.36 |
| **2018** | 3.19 | 0.89 | 12.91 |

*Unit = Number of defined daily doses per 1,000 inhabitants per day (DID).

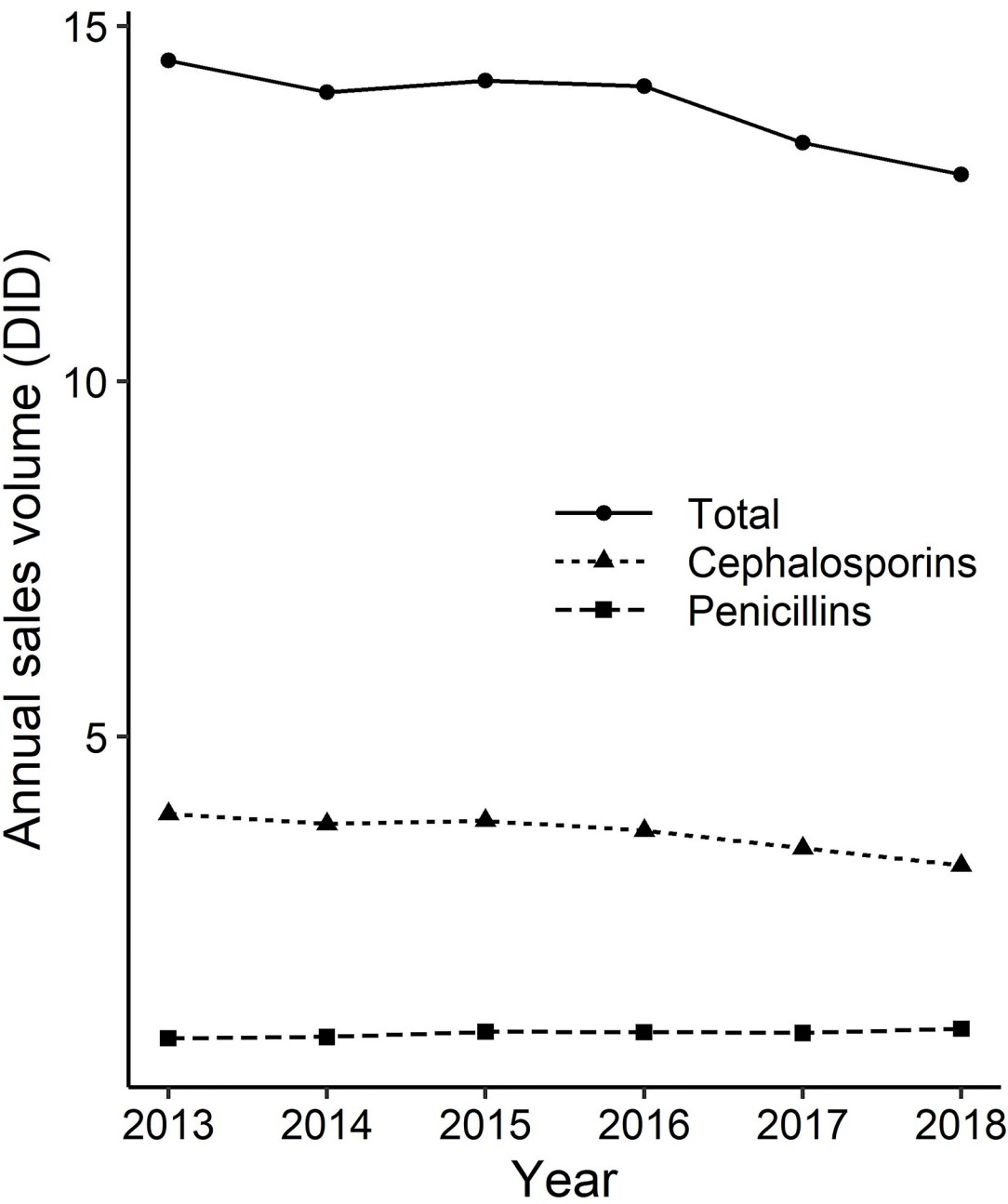

**Fig 5. Annual change in the sales volume of antibiotics.** The solid line with circles represents the total antibiotics sales, the dashed line with triangles represent the sales of cephalosporin, and the dashed line with squares represent the sales of penicillin. The vertical axis represents the number of defined daily doses per 1,000 inhabitants per day (DID).

### Correlation between susceptibility and antibiotic consumption

The sales volume of penicillins positively correlated with the rate of penicillin-susceptible isolates (Spearman's correlation coefficient: 0.801, $p < 0.001$). The sales volume of cephalosporins and total antibiotics negatively correlated with susceptibility (Spearman's correlation coefficient: -0.981 and -0.888, both $p < 0.001$). The details of the correlation tests are presented in Table 3.

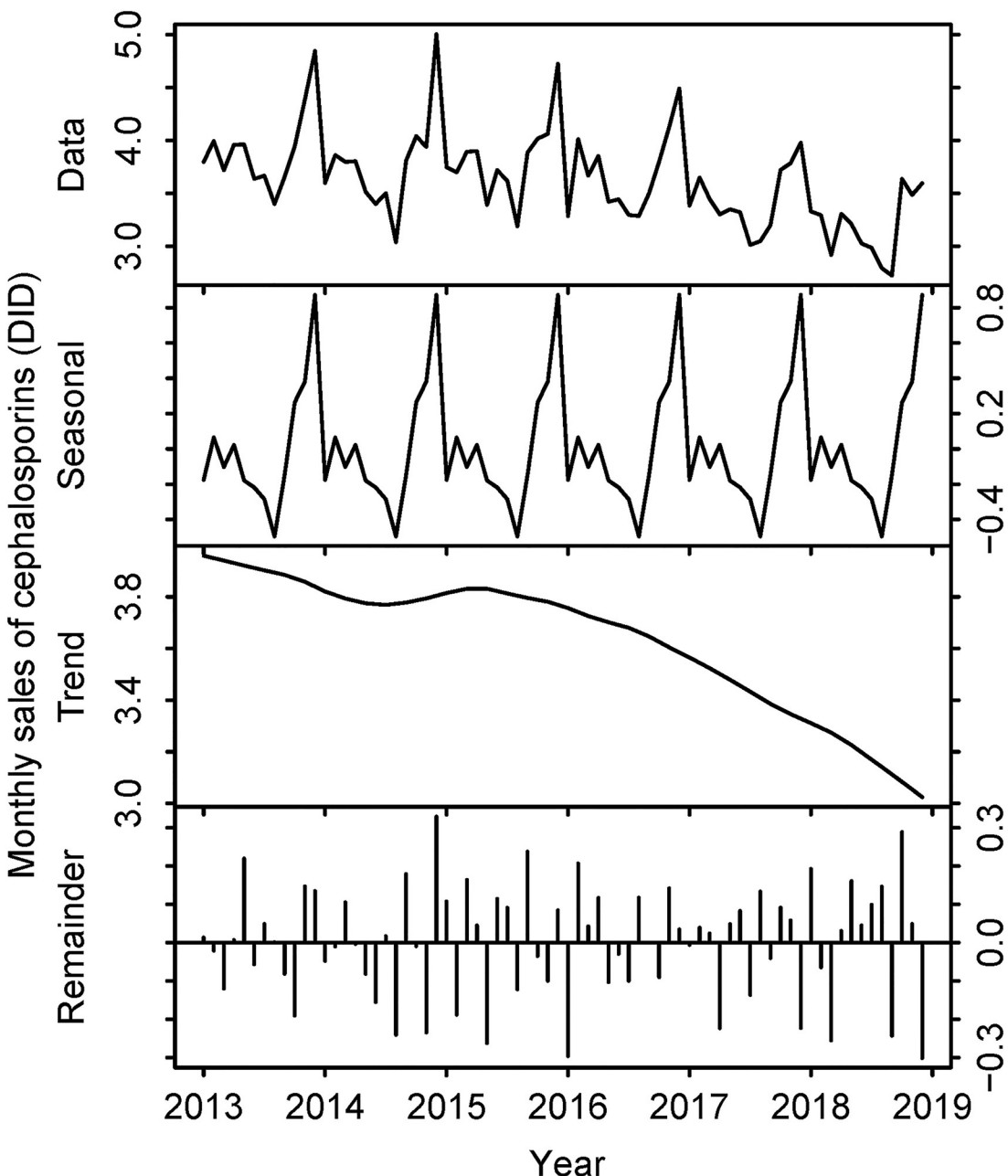

**Fig 6. Sales volume of cephalosporins in Japan, 2013–2018.** The top row represents the raw data. The second row describes seasonality components. The third row describes trend components. The bottom row represents the remainder. Horizontal axes represent month and year. Vertical axes represent the number of defined daily doses per 1,000 inhabitants per day (DID).

## Discussion

In this study, we evaluated CLSI meningitis MIC (or EUCAST criterion) as an indicator for the chronological trend of penicillin susceptibility of *S. pneumoniae*. Our results showed that meningitis MIC can provide a more intelligible way of penicillin susceptibility follow-up than that of non-meningitis MIC. As mentioned above, JANIS uses MIC values defined by CLSI,

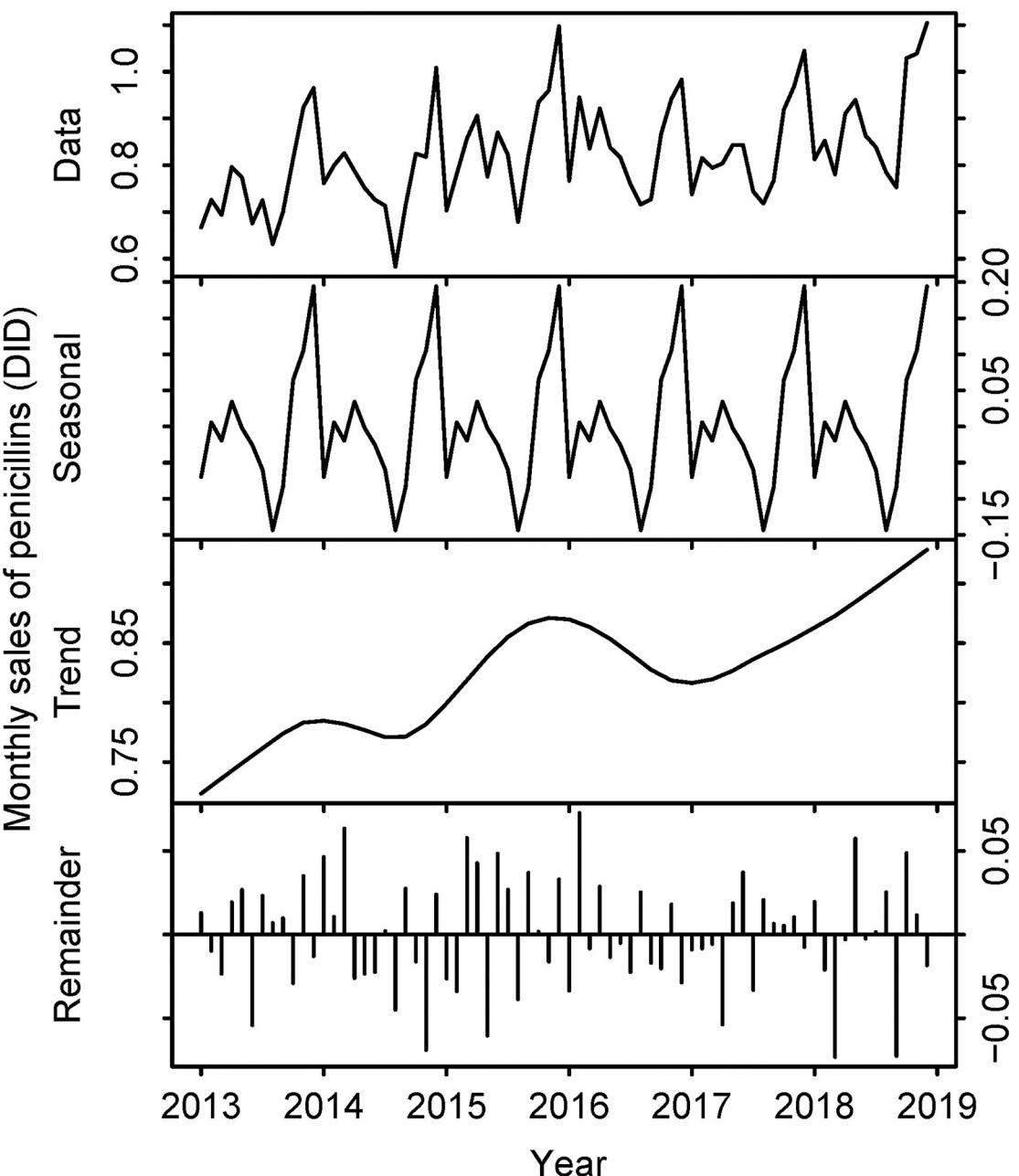

**Fig 7. Sales volume of penicillins in Japan, 2013–2018.** The top row represents the raw data. The second row describes seasonality components. The third row describes trend components. The bottom row represents the remainder. Horizontal axes represent month and year. Vertical axes represent the number of defined daily doses per 1,000 inhabitants per day (DID).

and non-meningitis MICs do not reflect the present results because most of the *S. pneumoniae* isolates were penicillin susceptible according to non-meningitis MICs.

We have already recognized the utility of CLSI non-meningitis MICs as a threshold for the clinical choice of antibiotics for preventing antibiotic abuse because most bacterial pneumoniae cases caused by *S. pneumoniae* can be treated by penicillin, even if the MICs are above 0.06 mg/L [26,27,37–39]. Nevertheless, meningitis MICs are beneficial for AMR surveillance; therefore, the rate of penicillin susceptibility of all *S. pneumoniae* isolates should be assessed

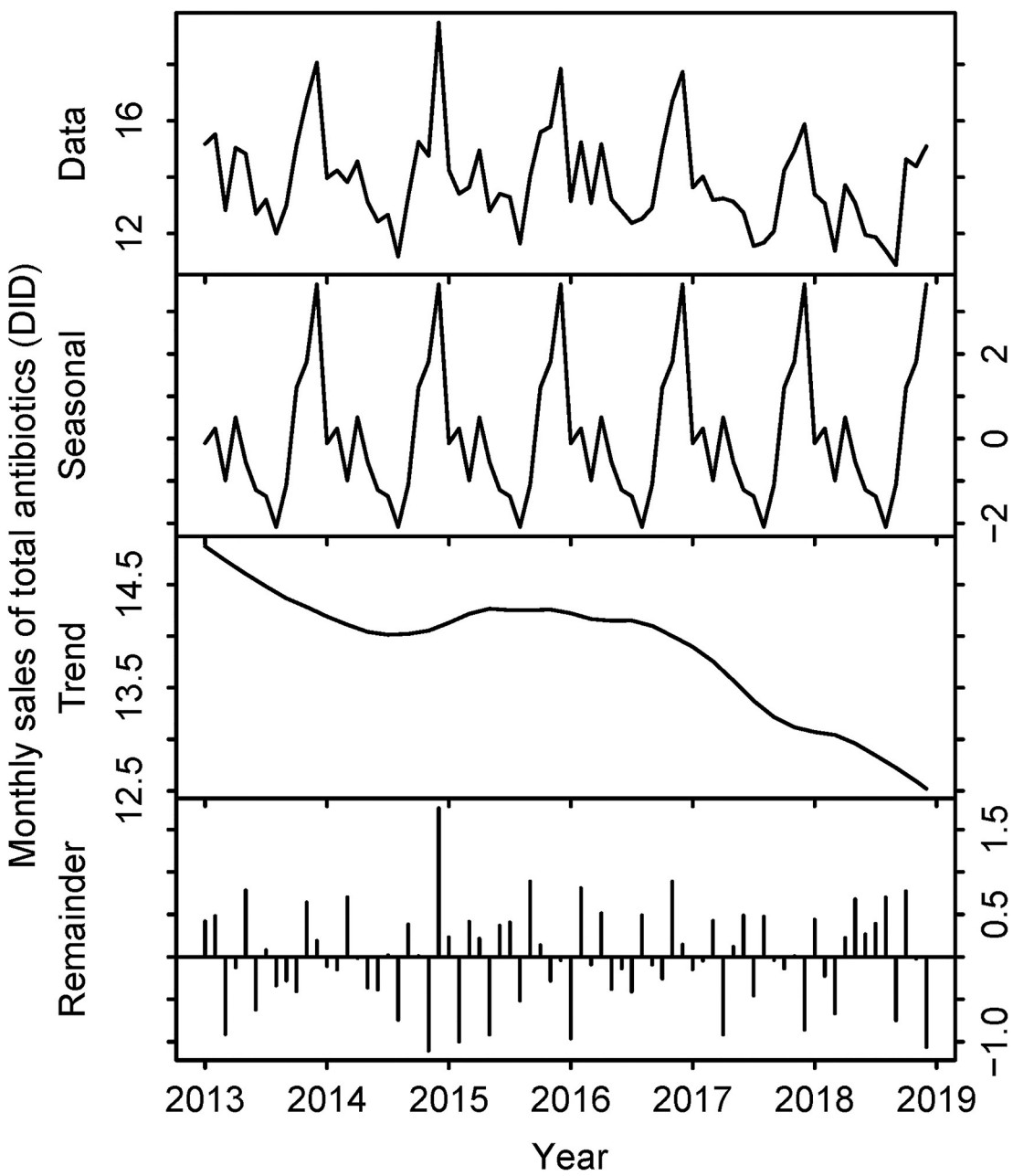

**Fig 8. Sales volume of total antibiotics in Japan, 2013–2018.** The top row represents the raw data. The second row describes seasonality components. The third row describes trend components. The bottom row represents the remainder. Horizontal axes represent month and year. Vertical axes represent the number of defined daily doses per 1,000 inhabitants per day (DID).

continuously by both non-meningitis and meningitis MICs. EUCAST criteria are useful if we regard 0.06 mg/L as the threshold for surveillance and 2.0 mg/L as the threshold for clinical use.

As for trend of antibiotic use, abuse of unnecessary antibiotics has been an important health issue in Japan, then various interventions have been implemented in Japanese healthcare system. For example, Ministry of Health, Labour and Welfare Japan became to provide incentives to healthcare facilities which implemented infection control team and other interventions in

**Table 3. Correlation between antibiotic consumption and rate of susceptible isolates, 2013–2018.**

|  | Coefficient* | *p*-value |
|---|---|---|
| **Cephalosporins** | -0.981 | < 0.001 |
| **Penicillins** | 0.801 | < 0.001 |
| **All antibiotics** | -0.888 | < 0.001 |

*Spearman's rank correlation coefficient.

Susceptible isolates were defined as those MICs ≤0.06 mg/L.

2012 [40]. Although publication of National Action Plan was 2016, such interventions were considered to have influence on antimicrobial consumption at national level.

Increase of penicillin consumption can be considered as a result of promoting appropriate antibiotic use because cephalosporin and other classes of antibiotic consumption in Japan was higher than that in other countries [41]. When we reduced inappropriate cephalosporin and other classes of antibiotic consumption, a part of it should be replaced by penicillin.

Another novel finding is that the increase in penicillin consumption is not negatively associated with an increase in penicillin susceptibility in *S. pneumoniae*. On the contrary, our results showed that the sales volume of penicillins positively correlated with the rate of penicillin-susceptible isolates.

Of course, this should not be interpreted directly because the National Action Plan on Antimicrobial Resistance encouraged not only the use of narrow-spectrum antibiotics like penicillins but also the reduction of unnecessary prescriptions of any class of antibiotics [11]. As a result, the sales of cephalosporins and the sum of all classes of antibiotics decreased gradually during the study period. Similarly, antibiotic use decreased during the same period [42,43]. The improved penicillin susceptibility rate of *S. pneumoniae* might be explained by this gradual decrease in total antibiotic consumption; hence, the positive correlation between the consumption of penicillins and susceptibility is not a causative relationship. Indeed, the negative correlation between total antibiotic consumption and penicillin susceptibility of *S. pneumoniae* is compatible with the findings of previous studies [44,45].

Considering these findings, it seems that we do not need to change our current recommendation, which is the use of penicillin as the first choice for the treatment of pneumococcal pneumonias and bloodstream infections, because it will be unlikely to induce penicillin resistance, and the use of total antibiotics and cephalosporins is more strongly associated with a decrease of penicillin-susceptible isolates. It is possible that even in case the increase of penicillin consumption negatively influences the susceptibility of *S. pneumoniae*, its influence might be cancelled by positive influence of consumption reduction of other classes. To encourage use "Access" drugs would be desirable in order to optimize antibiotic use at the population level [46–48].

Our data demonstrated obvious seasonality. Both sales data and penicillin resistance peaked in winter and were at their lowest levels in summer, similar to previous studies [44,45,49–51]. The correlation between the sales volume and susceptibility was strongest without lags, rather than with lags. Therefore, we are not sure about the length of time needed to observe the influence of changes in sales of antibiotics on susceptibility. It is easy to understand why antibiotic consumption peaks in winter, which is because acute respiratory tract infections are more commonly seen during this season; therefore, both appropriate and inappropriate use of antibiotics increases in winter. However, we are not sure why antibiotic resistance in *S. pneumoniae* peaks at the same time. This is another issue for further consideration.

This study had several limitations. First, our results showed correlation rather than causation. It is likely that increased penicillin consumption does not induce penicillin resistance in *S. pneumoniae* at the population level; however, it is also possible that improved penicillin susceptibility induces an increase in penicillin prescriptions. Second, we used only sales data and susceptibility data, and other factors (e.g. genetic mechanism) were not included in our analyses. Because various interventions, which include education for the general population, were implemented after (and even before) publishing the National Action Plan on Antimicrobial Resistance in Japan, social factors other than the decrease in antibiotic consumption might have had an influence on the susceptibility of *S. pneumoniae* [52–55]. If we can take other societal factors into consideration, we may be able to evaluate the influence of change in antibiotic consumption on susceptibility more precisely.

## Conclusions

Our findings suggest that meningitis MICs defined by CLSI might be useful indicators for the continuous monitoring of the susceptibility of *S. pneumoniae* to penicillins. The increase in penicillin use is not positively associated with susceptibility of *S. pneumoniae* isolates, which suggests that the current recommendations are reasonable. Further study including genetic mechanism of penicillin susceptibility of *S. pneumoniae* would reveal additional details of the relationship between antibiotic consumption and antibiotic resistance in *S. pneumoniae*.

## Supporting information

**S1 Data.**
(ZIP)

## Acknowledgments

We would like to thank all the facilities that participated in JANIS.

## Author Contributions

**Conceptualization:** Nobuaki Matsunaga, Motoyuki Sugai.

**Data curation:** Koji Yahara, Keigo Shibayama.

**Formal analysis:** Shinya Tsuzuki.

**Funding acquisition:** Norio Ohmagari.

**Investigation:** Shinya Tsuzuki, Nobuaki Matsunaga.

**Methodology:** Shinya Tsuzuki, Takayuki Akiyama.

**Project administration:** Shinya Tsuzuki, Nobuaki Matsunaga, Norio Ohmagari.

**Supervision:** Shinya Tsuzuki, Norio Ohmagari.

**Validation:** Shinya Tsuzuki.

**Visualization:** Takayuki Akiyama.

**Writing – original draft:** Shinya Tsuzuki.

**Writing – review & editing:** Takayuki Akiyama, Nobuaki Matsunaga, Koji Yahara, Keigo Shibayama, Motoyuki Sugai, Norio Ohmagari.

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
