## [Decision Letter · Decision Letter 0]

21 Aug 2020

PONE-D-20-23245

Improved penicillin susceptibility of Streptococcus pneumoniae and increased penicillin consumption in Japan, 2013–18

PLOS ONE

Dear Dr. Tsuzuki,

Thank you for submitting your manuscript to PLOS ONE. After careful consideration, we feel that it has merit but does not fully meet PLOS ONE’s publication criteria as it currently stands. Therefore, we invite you to submit a revised version of the manuscript that addresses the points raised during the review process.

We look forward to receiving your revised manuscript.

Kind regards,

Shampa Anupurba, MD

Academic Editor

PLOS ONE

Journal Requirements:

2.Thank you for stating the following in the Funding Section of your manuscript:

[This study was funded by a grant 256 from the Ministry of Health, Labour and

Welfare (Grant number H29 shinkougyousei shitei 005) and Research Program on

Emerging and Re-emerging Infectious Diseases from the Japan Agency for Medical

Research and Development (AMED) under grant number JP19fk0108061.]

 [The funders had no role in study design, data collection and analysis, decision to publish, or preparation of the manuscript.]

3.We note that you have indicated that data from this study are available upon request. PLOS only allows data to be available upon request if there are legal or ethical restrictions on sharing data publicly. For information on unacceptable data access restrictions, please see http://journals.plos.org/plosone/s/data-availability#loc-unacceptable-data-access-restrictions.

4. Your ethics statement must appear in the Methods section of your manuscript. If your ethics statement is written in any section besides the Methods, please move it to the Methods section and delete it from any other section. Please also ensure that your ethics statement is included in your manuscript, as the ethics section of your online submission will not be published alongside your manuscript.

Reviewers' comments:

Reviewer's Responses to Questions

**Comments to the Author**

1. Is the manuscript technically sound, and do the data support the conclusions?

Reviewer #1: Yes

Reviewer #2: Yes

2. Has the statistical analysis been performed appropriately and rigorously? 

Reviewer #1: Yes

Reviewer #2: Yes

3. Have the authors made all data underlying the findings in their manuscript fully available?

Reviewer #1: Yes

Reviewer #2: Yes

4. Is the manuscript presented in an intelligible fashion and written in standard English?

Reviewer #1: Yes

Reviewer #2: Yes

5. Review Comments to the Author

Reviewer #1: The authors have carried out an important data analysis to reflect on the Penicillin susceptibility of Streptococcus pneumoniae and to show its relationship with consumption . They have collated data from JANIS , collected from 2000 healthcare setups. Appropriate statistical tools have been applied to show the changing outcomes . The diagrammatic representations would of the useful if appropriate legends as and explanatory statements had been placed for the table.

Few queries that and be addressed by the authors :

Whether the changes across the years were significantly different for Penicillin?

Why did the antibiotic consumption come down across the spectrum?

Was any regulatory mechanism influence the decrease?

It is interesting to note that as consumption went up the resistance came down, contrary to existing belief.

This calls for further look at genetic mechanism and factors influencing

Reviewer #2: The manuscript conveys an important message that, used appropriately, penicillin consumption may not be necessarily associated with increased resistance of Streptococcus pneumoniae to penicillin, by analyzing penicillin consumption data with penicillin susceptibility data over 5 years. They also addressed the relative suitability of various MIC breakpoints for such surveillance. It is well written. The figures have an orientation problem, particularly for the text, are difficult to interpret and needs attention. Though the study has been based on MIC, it only presents categorical interpretation (susceptible or resistant) of Streptococcus pneumoniae and doesn't mention the actual MICs of the isolates anywhere. A scatter diagram showing the exact MICs of isolates over time is desirable and will enhance understanding.

6. PLOS authors have the option to publish the peer review history of their article (what does this mean?). If published, this will include your full peer review and any attached files.

Reviewer #1: **Yes: **Reba Kanungo

Reviewer #2: **Yes: **Dr Pallab Ray

---

## [Author Response · Author response to Decision Letter 0]

21 Sep 2020

18th September 2020

Shampa Anupurba,

Dear Dr Anupurba, 

Thank you very much for the comments regarding the manuscript we submitted, entitled “Improved penicillin susceptibility of Streptococcus pneumoniae and increased penicillin consumption in Japan, 2013–18” (ONE-D-20-23245), and for the opportunity to revise the paper. The reviewers’ and your feedback were great help in highlighting ways to improve our manuscript. Attached you will find a revised version of the manuscript with track changes, a cleaned version of the revision. Following this letter are reviewers’ comments with our responses in blue italics. 

We believe that the paper improved significantly, and we hope that you agree. Once again, thank you for the helpful feedback.

Sincerely, 

Shinya Tsuzuki, MD, MSc

AMR Clinical Reference Center, 

National Center for Global Health and Medicine

stsuzuki@hosp.ncgm.go.jp

ONE-D-20-23245

Improved penicillin susceptibility of Streptococcus pneumoniae and increased penicillin consumption in Japan, 2013–18

PLOS ONE

Journal Requirements:

2.Thank you for stating the following in the Funding Section of your manuscript:

[This study was funded by a grant 256 from the Ministry of Health, Labour and

Welfare (Grant number H29 shinkougyousei shitei 005) and Research Program on

Emerging and Re-emerging Infectious Diseases from the Japan Agency for Medical

Research and Development (AMED) under grant number JP19fk0108061.]

 [The funders had no role in study design, data collection and analysis, decision to publish, or preparation of the manuscript.]

We removed this sentence. Our study was supported by a grant from the Ministry of Health, Labour and Welfare (Grant number 20HA2003, changed from the previous one) and Research Program on Emerging and Re-emerging Infectious Diseases from the Japan Agency for Medical Research and Development (AMED) under grant number JP19fk0108061.]. The funders had no role in study design, data collection and analysis, decision to publish, or preparation of the manuscript. No author received a salary from the funders.

3.We note that you have indicated that data from this study are available upon request. PLOS only allows data to be available upon request if there are legal or ethical restrictions on sharing data publicly. For information on unacceptable data access restrictions, please see http://journals.plos.org/plosone/s/data-availability#loc-unacceptable-data-access-restrictions.

Thank you for your suggestion. We uploaded the data we used in the present study.

4. Your ethics statement must appear in the Methods section of your manuscript. If your ethics statement is written in any section besides the Methods, please move it to the Methods section and delete it from any other section. Please also ensure that your ethics statement is included in your manuscript, as the ethics section of your online submission will not be published alongside your manuscript.

Ethical approval is stated in “Data Source” subsection in “Materials and Methods” section.

Reviewer #1: The authors have carried out an important data analysis to reflect on the Penicillin susceptibility of Streptococcus pneumoniae and to show its relationship with consumption. They have collated data from JANIS, collected from 2000 healthcare setups. Appropriate statistical tools have been applied to show the changing outcomes. The diagrammatic representations would of the useful if appropriate legends as and explanatory statements had been placed for the table.

Thank you very much for your comment. We updated Figure 3 and 5 as an explanatory diagram for the contents of Table 1 and 2. We also made Figure 4 according to another reviewer’s suggestion, then original Figure 3, 4, and 5 were renamed as Figure 6, 7, and 8.

In addition, we modified the title and the legend of Table 3 to help readers’ understanding. We added the duration of the study (2013-2018) to the original title and a footnote about MICs used (isolates whose MICs ≤0.06 mg/L were classified as “susceptible”, i.e. we used “Meinigitis MIC” in Table 3).

Few queries that and be addressed by the authors:

Whether the changes across the years were significantly different for Penicillin?

Thank you for your question. When we conducted a linear regression analysis after decomposing seasonality influence, sales data of penicillins showed significant increase trend (p < 0.001). In contrast, total sales of antibiotics and cephalosporin showed decrease trend (p < 0.001).

Why did the antibiotic consumption come down across the spectrum?

Thank you for giving us an opportunity to explain this issue. These questions stated above can be explained by not only National Action Plan of our country (reference [11] and please also refer to our “Introduction” section) but also other countermeasures before publication of National Action Plan. Abuse of unnecessary antibiotics has been an important health issue in Japan, then various interventions have been implemented in Japanese healthcare system. For example, Ministry of Health, Labour and Welfare Japan became to provide incentives to healthcare facilities which implemented infection control team and other interventions in 2012. Although publication of National Action Plan was 2016, such interventions were considered to have influence on antimicrobial consumption across the spectrum at national level. 

As for penicillin consumption, it can be considered as a result of promoting appropriate antibiotic use because the use of cephalosporin and other classes of antibiotic consumption in Japan was higher than that in other countries. When we reduced inappropriate cephalosporin and other classes of antibiotic consumption, a part of it should be replaced by penicillin. These explanations are also added in “Discussion” section in the revised manuscript. 

Was any regulatory mechanism influence the decrease?

As explained in our answer to the previous question, our National Action Plan might be considered as one of important regulatory factor. In addition, we have implemented several interventions before publication of National Action Plan such as incentive provision for appropriate antibiotic use.

It is interesting to note that as consumption went up the resistance came down, contrary to existing belief.

This calls for further look at genetic mechanism and factors influencing

We appreciate your insightful comment. As stated in “Discussion” section, it is one of limitations that we did not include factors other than antibiotics sales data. Then we explicitly discussed in the main text that we did not take genetic mechanisms into consideration, and further research about mechanisms between penicillin susceptibility and its consumption will be a future challenge.

One of reasonable hypotheses is that, deeply related to our previous answer, reduction in other classes of antibiotics had positively influenced on penicillin susceptibility of S. pneumoniae. Especially, since cephalosporins are another class of beta-lactams, then consumption reduction in this class might do good to penicillin susceptibility. It is possible that even in case the increase of penicillin consumption negatively influences the susceptibility of S. pneumoniae, its influence might be cancelled by positive influence of consumption reduction of other classes. We also explained this hypothesis in “Discussion” section in the revised manuscript.

Reviewer #2: The manuscript conveys an important message that, used appropriately, penicillin consumption may not be necessarily associated with increased resistance of Streptococcus pneumoniae to penicillin, by analyzing penicillin consumption data with penicillin susceptibility data over 5 years. They also addressed the relative suitability of various MIC breakpoints for such surveillance. It is well written. The figures have an orientation problem, particularly for the text, are difficult to interpret and needs attention. Though the study has been based on MIC, it only presents categorical interpretation (susceptible or resistant) of Streptococcus pneumoniae and doesn't mention the actual MICs of the isolates anywhere. A scatter diagram showing the exact MICs of isolates over time is desirable and will enhance understanding.

Thank you very much for your constructive suggestion. We made a new cumulative bar chart as you can see the actual distribution of MICs of the isolates (Figure 4). We did not make a scatter diagram but a cumulative bar chart, because MICs take discrete values (e.g. 0.06, 0.12, 0.25, and so forth) then if we try to “scatter” them, they do not seem as a scattered plot. We believe that the bar chart makes it clear that the proportion of isolates whose MICs are ≤0.06 mg/L (“Susceptible”) has been increased gradually and isolates whose MICs are higher than 2 mg/L has not changed drastically, and it will help readers’ understanding. We also updated Figure 3 and 5 according to another reviewer’s suggestion, they explained the contents of Table 1 and 2 diagrammatically. Consequently, original Figure 3, 4, and 5 were renamed as Figure 6, 7, and 8.

---

## [Decision Letter · Decision Letter 1]

1 Oct 2020

Improved penicillin susceptibility of Streptococcus pneumoniae and increased penicillin consumption in Japan, 2013–18

PONE-D-20-23245R1

Dear Dr. Tsuzuki,

We’re pleased to inform you that your manuscript has been judged scientifically suitable for publication and will be formally accepted for publication once it meets all outstanding technical requirements.

Kind regards,

Shampa Anupurba, MD

Academic Editor

PLOS ONE

Reviewers' comments:

Reviewer's Responses to Questions

**Comments to the Author**

1. If the authors have adequately addressed your comments raised in a previous round of review and you feel that this manuscript is now acceptable for publication, you may indicate that here to bypass the “Comments to the Author” section, enter your conflict of interest statement in the “Confidential to Editor” section, and submit your "Accept" recommendation.

Reviewer #1: All comments have been addressed

Reviewer #2: All comments have been addressed

2. Is the manuscript technically sound, and do the data support the conclusions?

Reviewer #1: Yes

Reviewer #2: Yes

3. Has the statistical analysis been performed appropriately and rigorously? 

Reviewer #1: Yes

Reviewer #2: Yes

4. Have the authors made all data underlying the findings in their manuscript fully available?

Reviewer #1: Yes

Reviewer #2: Yes

5. Is the manuscript presented in an intelligible fashion and written in standard English?

Reviewer #1: Yes

Reviewer #2: Yes

6. Review Comments to the Author

Reviewer #1: The authors have explained and clarified the reviewers comments satisfactorily. The manuscript can now be accpted for publication.

Reviewer #2: The reviewer's comments have been adequately addressed to satisfaction and the manuscript is acceptable for publication.

7. PLOS authors have the option to publish the peer review history of their article (what does this mean?). If published, this will include your full peer review and any attached files.

Reviewer #1: **Yes: **Reba Kanungo

Reviewer #2: **Yes: **Dr Pallab Ray

---

## [Editor Report · Acceptance letter]

9 Oct 2020

PONE-D-20-23245R1 

Improved penicillin susceptibility of *Streptococcus pneumoniae* and increased penicillin consumption in Japan, 2013–18 

Dear Dr. Tsuzuki:

I'm pleased to inform you that your manuscript has been deemed suitable for publication in PLOS ONE. Congratulations! Your manuscript is now with our production department. 

Kind regards, 

on behalf of

Dr. Shampa Anupurba 

Academic Editor

PLOS ONE